# Exploring the effects of hypoxia and reoxygenation time on hepatocyte apoptosis and inflammation

Xinlu Xu[1,2☯], Tanfang Zhou[1☯], Alimu Tulahong[1], Rexiati Ruze[1], Yingmei Shao[1,2]*

1 Department of Hepatobiliary and Hydatid Disease, The First Affiliated Hospital of Xinjiang Medical University, Urumqi, China, 2 Key Laboratory of High Incidence Disease Research in Xinjiang (Xinjiang Medical University) Ministry of Education, Urumqi, China

☯ These authors contributed equally to this work.
* syingmei1@163.com

**Data Availability Statement:** All relevant data are within the paper and its Supporting Information files.

## Abstract

Hepatic Ischemia-Reperfusion Injury (HIRI) is an unavoidable pathological process during liver surgeries such as liver transplantation and hepatic resection, which involves a complex set of molecular and cellular mechanisms. The mechanisms of HIRI may involve a variety of biological processes in which inflammation and apoptosis play a central role. Therefore, it is crucial to deeply investigate the effects of different hypoxia and reoxygenation times on the construction of an *in vitro* model of hepatic ischemia-reperfusion injury. The human normal liver cell line HL-7702 IRI model was constructed by hypoxia chamber, and the inflammation and apoptosis focal levels of cells were detected by enzyme-linked immunosorbent assay, western blot and quantitative reverse transcription polymerase chain reaction. When 12-hour reoxygenation time was fixed, the inflammation and apoptosis indexes of HIRI model increased with the prolongation of hypoxia time (6, 12 and 24 hours). These indices reached highest level in the model group of 24-hour fixed hypoxia and 12-hour reoxygenation. Inflammation and apoptosis indices were significantly higher in the model group of 24-hours fixed hypoxia and 12-hours reoxygenation than in the group of 6 and 24 hours of reoxygenation. Taken together, the findings from this research demonstrated that during hypoxia phase, cells exhibited a clear time-dependent response of inflammation and cell death; on the contrary, during the reoxygenation phase, the cellular damage was not monotonically incremental, but showed an inverted U-shaped dynamic pattern. The present study reveals in depth the dynamic changes of cellular responses under hypoxia and reoxygenation conditions, providing us with an important theoretical basis to guide the selection and optimization of *in vitro* experimental models.

## Introduction

HIRI refers to a state of liver tissue where blood flow and oxygenation are temporarily halted, followed by the resumption of circulation and oxygen provision, resulting in a pathophysiologic process that aggravates hepatic injury and dramatically affects the recovery of patients'

**Funding:** the Open Project Fund for the State Key Laboratory of Central Asian High Disease Pathogenesis and Prevention (SKL-HIDCA-2023-2); the National Natural Science Foundation of China (82360111). The funder, represented by Yingmei Shao, has offered critical insights and constructive feedback that have been instrumental to our research. YS's contributions have significantly enhanced the manuscript by refining critical intellectual content, ensuring the study's integrity and elevating the academic quality of the paper.

**Competing interests:** The authors have declared that no competing interests exist.

postoperative liver function [1–3]. Someone pointed out that different characters of ischemia, including type,(cold or thermal ischemia), severity (partial or complete), duration (as short as a few minutes and as long as a few hours), as well as regenerative process (primarily when hepatic resection is performed), may influence mechanism of HIRI and regenerative disorders [4].

Hepatocytes and sinusoidal endothelial cells are the two hepatocyte types affected by HIRI most. Still, they show different sensitivities to different types of ischemia: hepatocytes are more sensitive to hot ischemia [5, 6], whereas LSEC is more sensitive to cold ischemia. study suggests that reperfusion after 120–180 minutes of thermal ischemia leads to irreversible cellular damage, and 90 minutes of thermal ischemia is the estimated time limit for hepatocyte survival [7]. Hypoxia duration and reoxygenation time are the main influencing factors leading to IRI in hepatocytes. But at present, the exact effects of the duration of hypoxia and reoxygenation on the inflammatory response and apoptotic processes in cells are not fully understood.

At human level, clinical studies are usually performed during liver surgery (e.g., hepatic resection, liver transplantation) to simulate HIRI by controlling the blockage and restoration of blood flow. Investigators have assessed the severity of HIRI and the recovery of the patients by monitoring the liver function indices and other relevant parameters before and after the surgery [8, 9]. At animal model level, various approaches to constructing HIRI involve different degrees of blood flow blockade and durations of ischemia and reperfusion. For example, commonly used models may include 70% blood flow blockade [10–12], followed by an one-hour blood supply interruption and a six-hour blood flow restoration [9]. This setup helps to simulate the clinical situation and assess the degree of injury. *In vitro* models at the cellular level are more variable, in which the duration of hypoxia ranges from 4 to 24 hours, and reoxygenation has several options from 4 to 24 hours [13, 14]. This diversity reflects the exploration of different injury stages and cellular stress responses. The study of experimental models of HIRI is essential for a more profound comprehension of molecular processes underlying liver injury, development of new therapeutic strategies, and improvement of patients' clinical prognosis. These models allow us to systematically investigate HIRI under strictly controlled conditions, revealing impact of hypoxia and reoxygenation durations on cellular inflammation and apoptosis, and providing robust scientific evidence for clinical treatment.

In this paper, we constructed a hepatocyte HIRI model by using hypoxia chamber method and selecting different hypoxia-reoxygenation times. Validation and comparison of the effects of constructing hepatocyte HIRI in the above ways in terms of inflammation and apoptosis, provide new insights into understanding how cells adjust their physiological state under hypoxic and reoxygenation conditions and deepened our understanding of the cell death pathway.

## Materials and methods

### Cell culture

The HL-7702, a typical human liver cell line, was preserved in the lab, revived, and cultivated in a complete 1640 medium which was enriched with a 10% addition of fetal bovine serum (Vivacell, Shanghai, China), along with 100 μg/mL of penicillin and 100 μg/mL of streptomycin (Vivacell, Shanghai, China), and the number of generations of cells used in the experiment varied from 10 to 15 at the beginning of the experiment. The cells were kept in a culture incubator with temperature of 37°C and $CO_2$ concentration of 5%.

### Modeling of hypoxia/reoxygenation

An *in vitro* hypoxia-reoxygenation model simulates typical human hepatocyte injury under reperfusion injury conditions. The complete 1640 culture medium enriched with 10% fetal

bovine serum (FBS) of HL-7702 cells was exchanged for a simple 1640 medium (Vivacell, Shanghai, China) and then placed into a hypoxic chamber (MIC-101, Billups-Rothenberg, USA). We precisely mixed 94% $N_2$, 1% $O_2$ and 5% $CO_2$ gases, calibrated by a flow meter to ensure the stability and homogeneity of the gas mixture. A highly sensitive oxygen sensor was installed in the hypoxia chamber to monitor the oxygen concentration in real-time. This monitoring step was critical to verify the oxygen concentration inside the chambers dropped to the hypoxic level required for the experiment. In this study, the hypoxic environment was maintained in the cell culture chamber for 6, 12, and 24 hours, after which the simple 1640 medium was changed to complete medium to establish the reoxygenation model. This study maintained the reoxygenated environment for 6, 12, and 24 hours. Reperfusion injury of the microcirculation was simulated *in vitro* by establishing a hypoxia-reoxygenation model.

## Western blotting to detect relevant antibodies

Cells of each group were collected, and total cellular protein was extracted. 2 μl of the supernatant was taken to measure the protein content. BCA was used to calculate the protein concentration. The rest was added to a 5×sampling buffer, and the metal bath denatured the protein at 100˚C for 10 min. Take 15 μl of protein, use 10% SDS polyacrylamide gel (Biotides, Beijing, China), and perform electrophoresis to separate the protein. Transferred to a PVDF membrane (Millipore, Billerica, MA, USA), skimmed milk was enclosed for 2 h. TNF-α (1:1000) (Abcam, Cambridge, UK), IL-1β (1:1000) (Abcam, Cambridge, UK), IL-6 (1:1000) (Bioss, Beijing, China), Caspase3 (1:500) (Proteintech, Wuhan, China), Caspase1 (1:2000) (Proteintech, Wuhan, China), Bcl2 (1:500) (Proteintech, Wuhan, China), GAPDH (1:5000) (Servicebio, Wuhan, China), β-actin (1:5000) (Proteintech, Wuhan, China) primary antibody, and placed in 4˚C overnight. The next day, the secondary antibody (Proteintech, Wuhan, China) affixed with HRP was added and incubated for 1.5 h. Exposure development was performed. ImageJ software determined the protein gray value, and the comparative protein levels in the remaining groups were determined by normalizing the protein levels of each group to their respective internal reference standards.

## Quantitative reverse transcription polymerase chain reaction

Total cellular RNA was isolated by using Trizol. Hepatic mRNA was converted to cDNA by using HiScript III RT SuperMix for qPCR (which includes gDNA removal) kit from Vazyme in Nanjing, China, following the protocol provided by the manufacturer. qRT-PCR was conducted by utilizing SYBR detection kit from Vazyme, Nanjing, China, compatible with the qRT-PCR system (Applied Biosystems, USA). Forward and reverse primers (Table 1) synthesized by Sangon (Shanghai, China) were used on a CFX96 real-time PCR system (Bio-Rad) by using a SsoFast EvaGreen Supermix (Bio-Rad, Foster City, CA) for real-time qPCR. GAPDH (Sangon, Shanghai, China) was used as an internal standardized reference.

## Enzyme-linked immunosorbent assay (ELISA) and liver enzyme assays

Cell supernatants were collected. TNF-α (Proteintech, Wuhan, China), IL-1β (CUSABIO, Wuhan, China), Bcl2(CUSABIO, Wuhan, China), Caspase3 (Proteintech, Wuhan, China) indexes, and alanine aminotransferase (ALT) as well as aspartate aminotransferase (AST) concentrations were determined by ELISA kits. All experiments were performed based on instructions.

**Table 1. Quantitative PCR primers.**

| Primers name | | Primers sequences (5'to3') | Length(bp) |
|---|---|---|---|
| TNF-α | Forward | CCTGACATCTGGAATCTGGAGACC | 24 |
| | Reverse | CTGGAAACATCTGGAGAGAGGAAGG | 25 |
| IL-1β | Forward | CAGTGGCAATGAGGATGACTTGTTC | 25 |
| | Reverse | CTGTAGTGGTGGTCGGAGATTCG | 23 |
| IL-6 | Forward | TGGTGTTGCCTGCTGCCTTC | 20 |
| | Reverse | GCTGAGATGCCGTCGAGGATG | 21 |
| Caspase3 | Forward | ATGGTTTGAGCCTGAGCAGAGAC | 23 |
| | Reverse | CGCCCTGGCAGCATCATCC | 19 |
| Bcl2 | Forward | CCGCATCAGGAAGGCTAGAGTTAC | 24 |
| | Reverse | GCTGGGACACAGGCAGGTTC | 20 |
| Caspase1 | Forward | CATCCCACCAGATACCTCCCATAAC | 25 |
| | Reverse | CTCTCCTCCCTTCTTGTGTGACTG | 24 |
| GAPDH | Forward | CACCCACTCCTCCACCTTTGAC | 22 |
| | Reverse | GTCCACCACCCTGTTGCTGTAG | 22 |

## Data analysis

Statistical comparisons across all datasets were conducted by one-way ANOVA function in SPSS, and the thresholds for the significance of the statistics were judged by the criteria (* for P values below 0.05, ** for P values below 0.01, *** for P values below 0.001, and **** for P values below 0.0001). GraphPad Prism 9.0, a software product of GraphPad Software from San Diego, California, was utilized for all statistical computations.

## Results

### 1. Modifications in hypoxia and reoxygenation durations induce varied degrees of cellular injury

We initially established a cellular model of HIRI with success and confirmed its scientific accuracy through careful examination of cell morphology and evaluation of hepatic enzyme functionality. Under the microscope, we recorded the changes in cell morphology in detail and observed that the cell injury showed a gradual aggravation with the prolongation of hypoxia time, especially at the time point of 24-hourhypoxia and 12-hour reoxygenation. Cell damage was the most obvious manifestation, with damaged cells showing shrinkage, reduced cell volume, and irregular edges. In addition, increased membrane vesicle formation was observed in the cytoplasm, which may be caused by endoplasmic reticulum stress and organelle dysfunction. Accumulation of cellular debris was also observed in our experiments, which may be a result of the release of cellular contents during apoptosis or necrosis (Fig 1A–1D, 1G–1J). In addition, activities of liver function-related enzymes, such as alanine aminotransferase (ALT) (Fig 1E and 1K) and aspartate aminotransferase (AST) (Fig 1F and 1L), were significantly elevated, further confirming the severity of cellular damage.

### 2. Inflammatory marker levels are directly proportional to the duration of hypoxia

After successful establishment of a cellular HIRI model, we analyzed the effects of hypoxia duration on hepatocytes. Firstly, we investigated how hypoxic environments influence the levels of inflammatory markers within cells, especially the changes in tumor necrosis factor-α

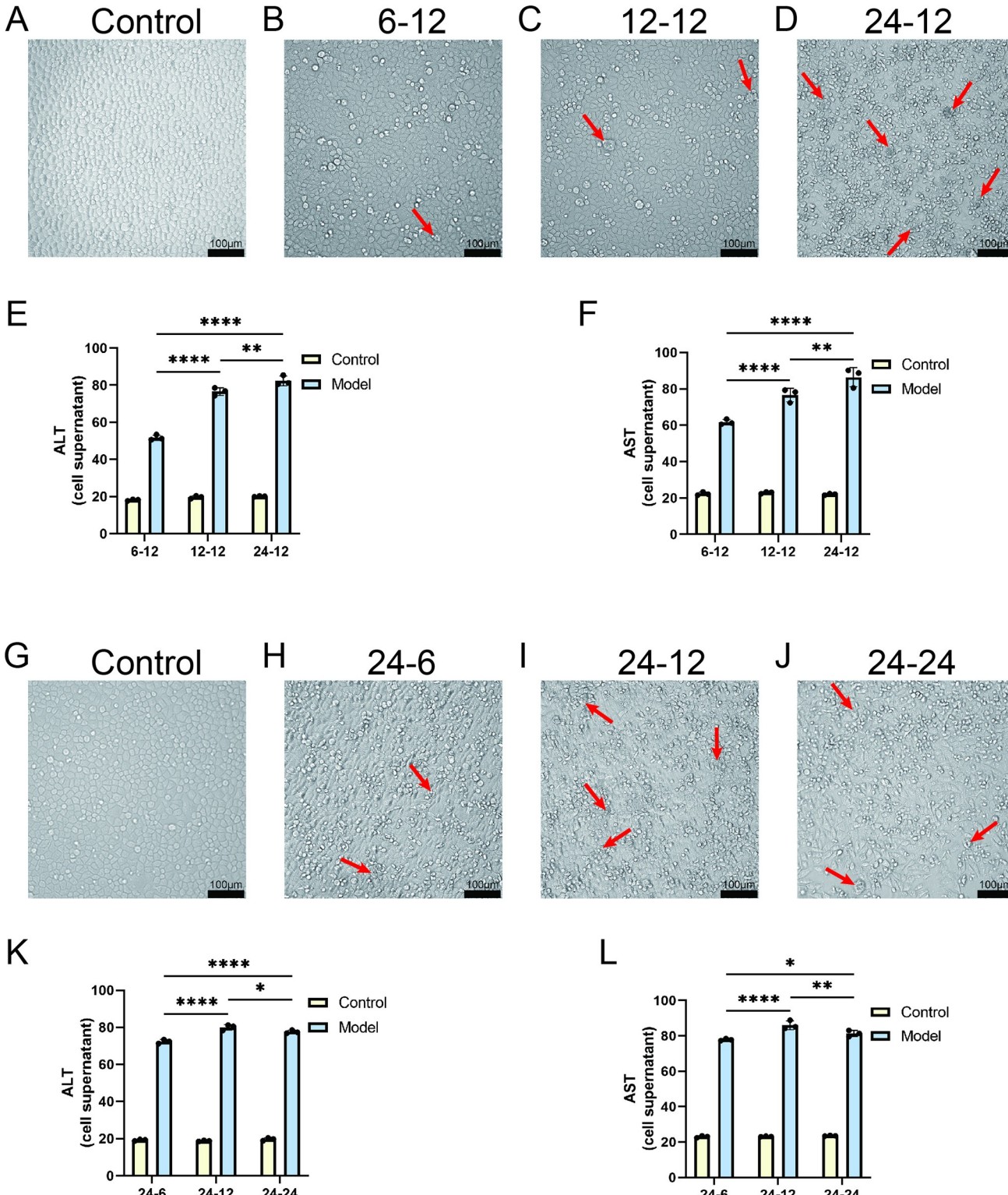

**Fig 1. Establishment and verification of cell IRI model.** Cellular changes observed under hypoxic conditions for 6h, 12h, 24h followed by reoxygenation for 6h, 12h, 24h. (Red arrows indicate an increase in vesicle formation and the accumulation of cellular debris, indicative of cellular damage and impaired membrane integrity) (A-D, G-J). ALT (E&K) and AST (F&L) levels in cells subjected to varying durations of hypoxia and reoxygenation (n = 3). Data are depicted as mean values accompanied by standard error of the mean (SEM), and the significance of the statistics is judged by the criteria (* for P values below 0.05, ** for P values below 0.01, *** for P values below 0.001, and **** for P values below 0.0001). Scale bar: 100 μm.

(TNF-α) and interleukin 1β (IL-1β). By quantitatively analyzing inflammatory mediators in cell supernatants, we observed that the levels of TNF-α and IL-1β increased significantly with the prolongation of hypoxia. This phenomenon was particularly pronounced from 0 to 24 hours of hypoxia (Fig 2A and 2B). Furthermore, we quantified the levels of TNF-α, IL-1β, and IL-6 proteins by using protein blotting. (Fig 2C). The experimental data showed that protein synthesis of these inflammatory factors significantly increased in hypoxic environments and this increase was proportional to the prolongation of hypoxia (Fig 2D–2F). By relative quantitative RT-PCR analysis, we found mRNA levels of inflammatory markers were consistent with protein expression trend, which gradually increased with the increase of hypoxia time (Fig 2G–2I). These data suggested that hypoxia promotes the protein synthesis of inflammatory factors and upregulates their mRNA levels, revealing a close link between cellular damage and inflammatory response.

## 3. Hypoxia induces apoptosis and pyroptosis

Next, we explored the effects of a hypoxic environment on apoptosis and pyroptosis by analyzing apoptosis-related indicators in cells. The levels of Bcl2 and Caspase3, essential indicators for assessing hypoxia-induces apoptosis, were examined. findings from the experiment indicated there was a significant increase in the concentrations of Bcl2 and Caspase3 as the duration of hypoxia was extended. There is a positive correlation ship between Caspase3 and Bcl2 concentration and hypoxia time (Fig 3A and 3B). From 0 to 24 h of hypoxia treatment, the expression levels of apoptosis markers increased incrementally along with time, reflecting the cumulative effect of cell injury. Using protein blotting technology, we quantified the protein expression of Bcl2, Caspase3, and Caspase1 (Fig 3C). The results showed that the synthesis of these apoptosis- and pyroptosis-related proteins was significantly increased under hypoxic conditions, and this increase was proportional to the duration of hypoxia (Fig 3D–3F). Moreover, we analyzed the mRNA expression of the corresponding genes by quantitative RT-PCR. The results revealed that the mRNA levels of apoptosis and pyroptosis-related genes gradually increased with the prolongation of hypoxia duration (Fig 3G–3I), which was consistent with the trend of protein levels. These results imply that hypoxia potentially plays a role in the molecular pathology of cellular damage by modulating the levels of proteins associated with apoptosis and pyroptosis.

## 4. Cellular inflammatory markers exhibited an inverted U-shaped trend during reoxygenation

After analyzing the hypoxia time, we started to study the reoxygenation time. First of all, we aimed to investigate the expression patterns of cellular inflammatory markers during reoxygenation, focusing on changes in tumor necrosis factor α (TNF-α) and interleukin 1β (IL-1β). Quantitative analysis showed that during reoxygenation, the levels of the cytokines TNF-α and IL-1β peaked at 12 h and then decreased at 24 h, revealing a time-dependent fluctuation of reoxygenation-induced inflammatory responses (Fig 4A and 4B). Using the protein blotting technique, we quantified the levels of TNF-α, IL-1β, and IL-6 proteins (Fig 4C). The data showed that the expression of these inflammatory factors showed significant dynamic fluctuations under reoxygenation conditions, which were closely related to the prolongation of reoxygenation time (Fig 4D–4F). Relative quantitative RT-PCR analysis showed that the mRNA levels of the inflammatory markers mirrored the pattern observed in protein expression levels, both of which peaked at 12 hours of reoxygenation and exhibited down-regulation at 24 hours, which further confirmed the dynamic regulatory nature of the inflammatory response (Fig 4G–4I). Our findings reveal the complexity of inflammatory marker expression during

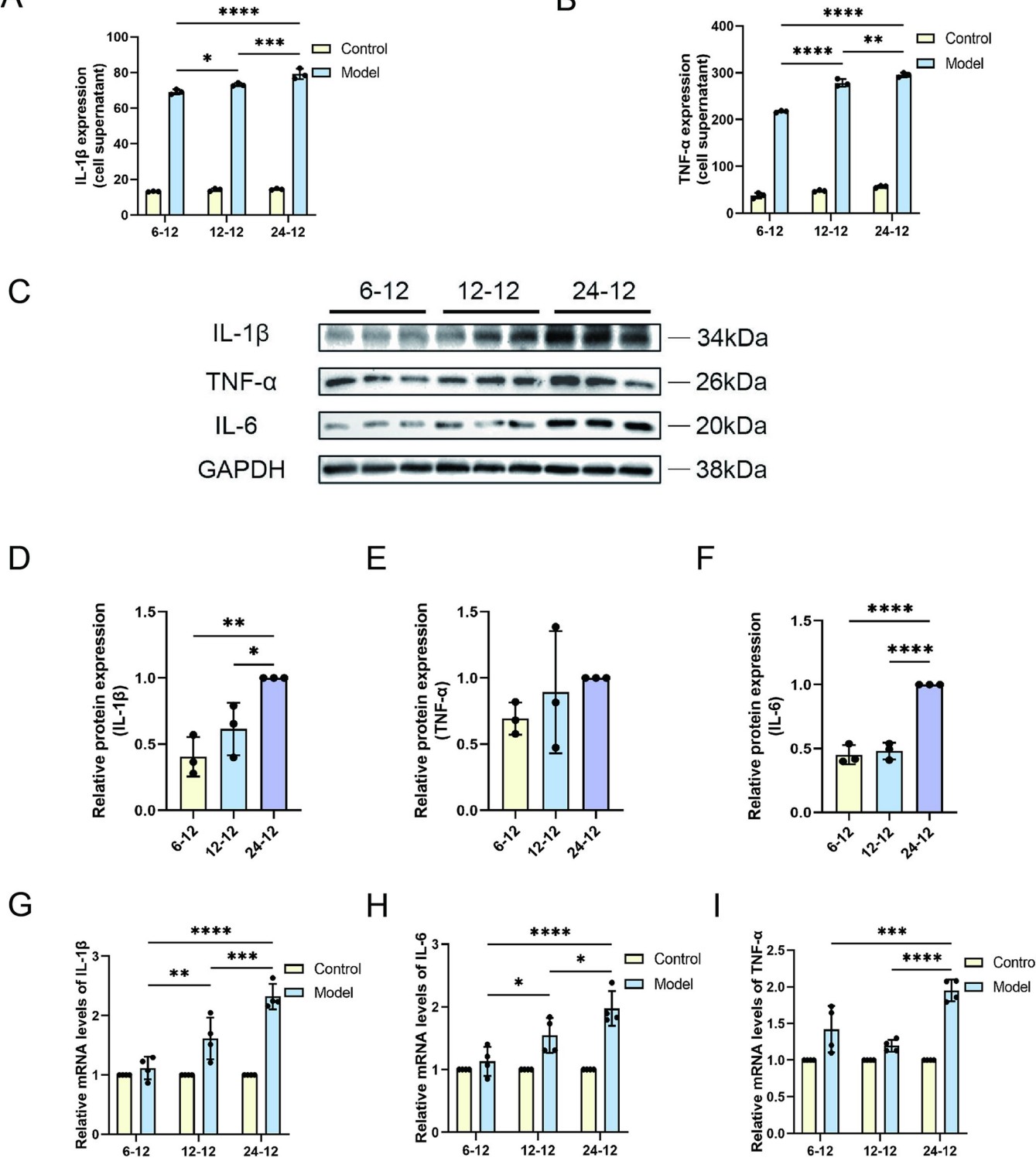

**Fig 2. Expression levels of inflammatory markers in response to hypoxia.** The quantities of expression for (A) IL-1β and (B) TNF-α regulators in cells via ELISA (n = 3). (C) Western blots measured IL-1β, TNF-α, IL-6 and GAPDH in HL-7702 cells. Relative protein expression of (D) IL-1β, (E) TNF-α, (F) IL-6 were quantified with the 24-hour hypoxia and 12-hour reoxygenation group (24–12) as a comparative benchmark for other experimental groups (n = 3). Quantitative real-time PCR (qRT-PCR) was conducted to analyze the (G) IL-1β, (H) IL-6, (I) TNF-α levels of cytokine within the cells (n = 4). Data are depicted as mean values accompanied by SEM, and the significance of the statistics is judged by the criteria (* for P values below 0.05, ** for P values below 0.01, *** for P values below 0.001, and **** for P values below 0.0001).

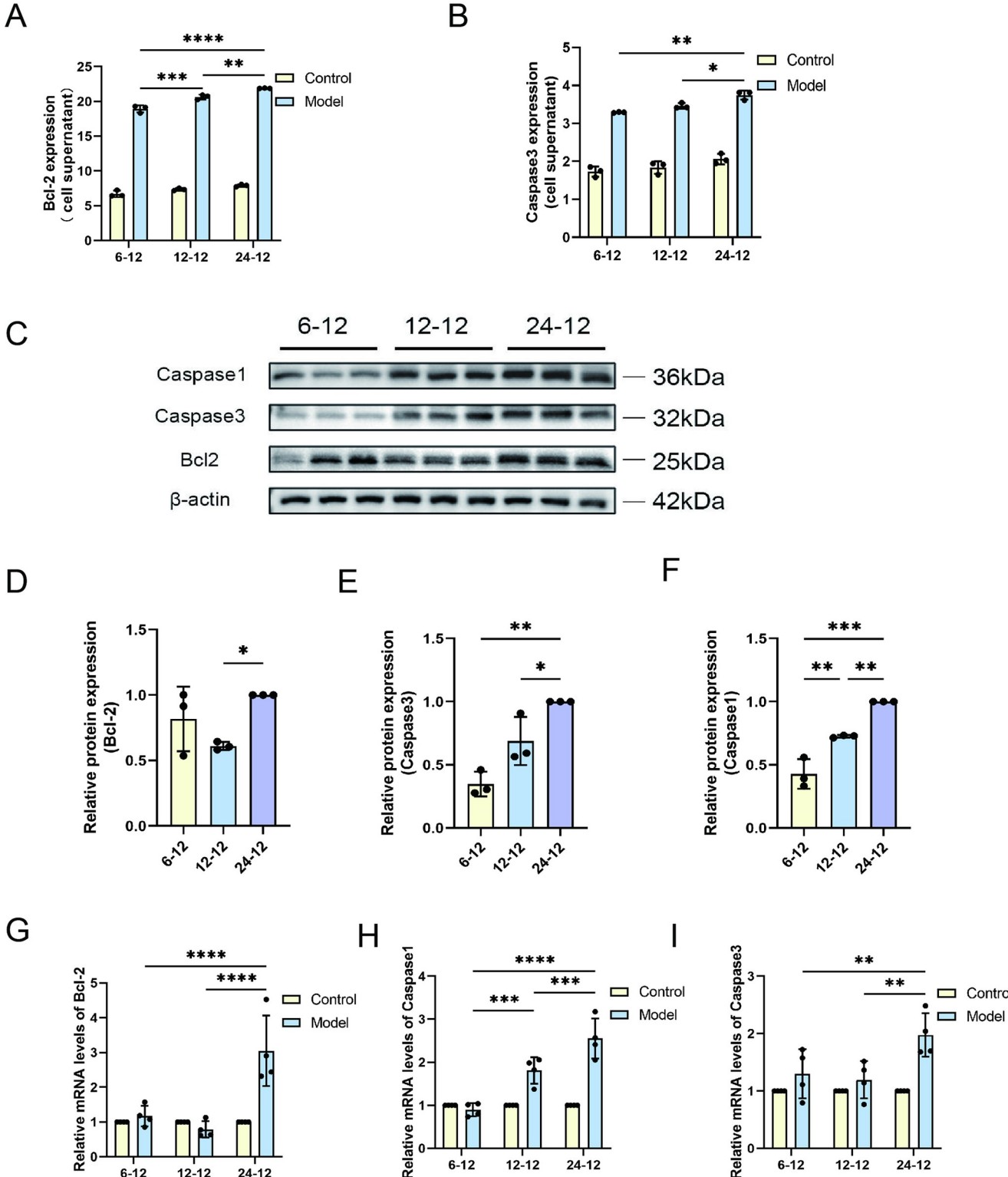

**Fig 3. Analysis of apoptosis and pyroptosis marker expression in response to different hypoxic time.** (A) Bcl2 and (B) Caspase3 levels were measured in cells using ELISA, with each condition assessed in triplicate (n = 3). (C) Protein expression levels of Bcl2, Caspase3, and Caspase1 were evaluated utilizing the technique of Western blotting, and adjusted in relation to the internal control gene β-actin within the cells. The comparative levels of (D) Bcl2, (E) Caspase3, and (F) Caspase1 proteins was quantified by normalizing to the 24-hour hypoxia and 12-hour reoxygenation group (24–12), serving as a comparative benchmark (n = 3). QRT-PCR was utilized to evaluate the mRNA levels of (G) Bcl2, (H) Caspase1, and (I) Caspase3 (n = 4). Data are depicted

as mean values accompanied by SEM, and the significance of the statistics is judged by the criteria (* for P values below 0.05, ** for P values below 0.01, *** for P values below 0.001, and **** for P values below 0.0001).

reoxygenation, suggesting that reoxygenation may trigger a series of molecular mechanisms that regulate the inflammatory response. These results enrich our understanding of the mechanism of reoxygenation-induced inflammatory response and provide new perspectives for developing therapeutic strategies against inflammatory diseases.

## 5. During reoxygenation, apoptosis and pyroptosis activities peaked then declined

In addition, we delved into the dynamics of apoptosis and pyroptosis during the reoxygenation phase. The significant effects of reoxygenation on cell death pathways through a series of experiments were revealed. Using the ELISA method, we tracked the concentrations of Bcl2 and Caspase3, identifying a peak in these critical markers at the 12-hour mark of reoxygenation, which was then followed by a decline in their expression levels at the 24-hour point (Fig 5A and 5B). This indicated a significant time-dependent fluctuation of the reoxygenation-induced apoptotic response. Using protein blotting technology, we further analyzed the expression patterns of Bcl2, Caspase3, and Caspase1 proteins (Fig 5C). The results showed that the protein levels of these indicators showed significant dynamic fluctuations under reoxygenation conditions, which were closely correlated with the prolongation of reoxygenation time (Fig 5D–5F). Relative quantitative RT-PCR analysis showed that the mRNA levels of apoptosis- and pyroptosis-related genes were consistent with the protein expression trend, which peaked at 12 h of reoxygenation and then showed a down-regulation at 24 h, which further confirmed the dynamic regulatory properties of apoptosis and pyroptosis responses (Fig 5G–5I). Our findings reveal the complexity of apoptosis and pyroptosis expression during reoxygenation and suggest that reoxygenation may activate a series of molecular mechanisms that regulate cell death pathways. These results enhance our understanding of the mechanisms of reoxygenation-induced apoptosis and pyroptosis responses and may be potentially valuable for controlling cell death and promoting tissue repair.

## Discussion

HIRI, is the damage that happens to the liver once its blood supply resumes following a period where it was temporarily cut off and is commonly seen during liver surgeries such as liver transplantation and hepatic resection [15, 16]. The main manifestations of HIRI injury are inflammation, apoptosis, and cell death. Inflammatory and apoptotic processes may be initiated during ischemia, while reperfusion may further accelerate these pathological events [17, 18]. Regulation of apoptosis and inflammation plays a crucial role in enhancing hepatocyte survival and reducing reperfusion injury [19, 20]. The inflammatory response activates immune cells, including Kupffer cells, which further release inflammatory factors during the reperfusion phase, such as TNF-α, IL-1β, and IL-6, which exacerbate inflammatory injury [14, 21]. In addition, the inflammatory response promotes apoptosis and necrosis of hepatocytes, and these forms of cell death have essential implications for the impairment of liver function [22]. During hepatic ischemia-reperfusion injury, the ischemic phase may stimulate apoptosis, a controlled, self-limiting process of cellular self-destruction that helps to eliminate damaged cells. Pyroptosis is another form of programmed cell death, triggered by the activation of inflammatory blisters, and is often associated with severe infection or tissue damage [23]. In the context of HIRI, focal death may occur as a result of the triggering of inflammatory

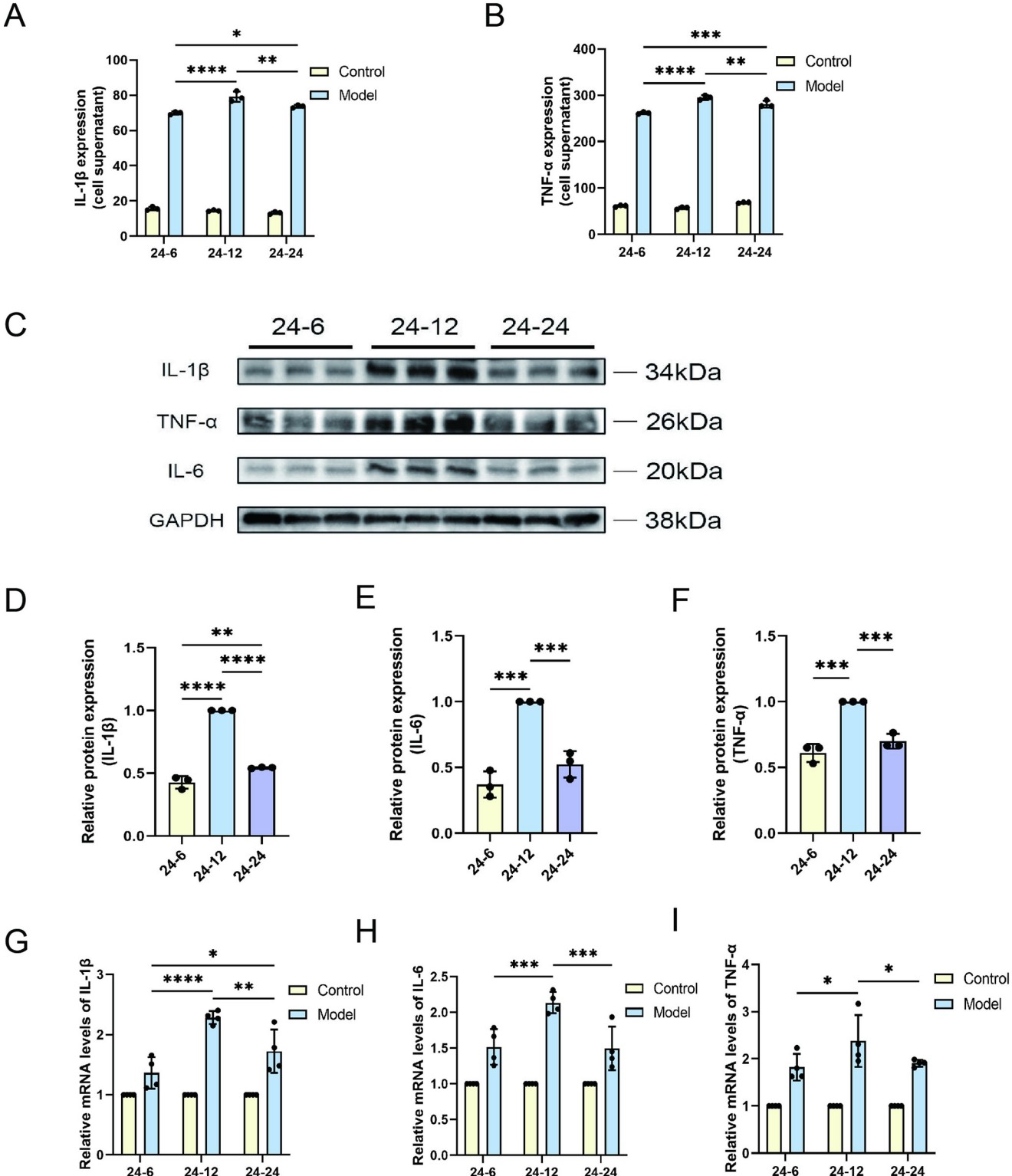

**Fig 4. Expression levels of inflammatory markers following varying durations of reoxygenation post-hypoxia in HL-7702 cells.** (A) IL-1β and (B) TNF-α levels in cells were determined using ELISA, with experiments conducted in triplicate (n = 3) to ensure reproducibility. (C) Protein expression levels of IL-1β, TNF-α, and IL-6 were evaluated utilizing the technique of Western blotting, and adjusted in relation to the internal control gene GAPDH. The comparative expression levels of the proteins (D) IL-1β, (E) IL-6, and (F) TNF-α were quantified and compared to the 24-hour hypoxia followed by 12-hour reoxygenation group (24–12), which served as the comparative benchmark. QRT-PCR was employed to analyze the mRNA levels expressed by

genes encoding (G) IL-1β, (H) IL-6, and (I) TNF-α within the cells, with normalization to the same benchmark group. Data are depicted as mean values accompanied by SEM, and the significance of the statistics is judged by the criteria (* for P values below 0.05, ** for P values below 0.01, *** for P values below 0.001, and **** for P values below 0.0001).

processes, which may further exacerbate inflammatory injury in the liver. Moreover, apoptosis and pyroptosis may have a cascading effect in HIRI, where clearance of dead cells and release of inflammatory mediators may activate neighboring immune cells, triggering more widespread inflammation and injury [24]. Thus, apoptosis and play critical roles in HIRI, and they affect the liver's response to injury and its ability to repair through diverse mechanisms.

In the process of HIRI, key factors such as oxidative stress, autophagy, mitochondrial dysfunction, inflammation, and apoptosis play critical roles. Oxidative stress is a complex process in HIRI involving restriction of blood supply, subsequent recovery and reoxygenation, leading to metabolic imbalance and oxidative damage. This process interacts with inflammation and oxidative damage to activate innate and adaptive immune responses, leading to cellular damage and organ dysfunction [1]. Mitochondrial autophagy, an intracellular clearance mechanism, selectively removes damaged mitochondria via autophagy pathway, preventing the release of cytochrome c and activation of the mitochondrial death pathway. In HIRI, mitochondrial autophagy is essential for maintaining the stability of mitochondrial function and cell survival. If not repaired promptly, mitochondrial dysfunction will lead to cellular energy metabolism disorders and cell death.

The liver consists of two major cell types, parenchymal and nonparenchymal cells, which together maintain complex functions of the liver. Parenchymal and nonparenchymal cells play different roles in hepatic ischemia-reperfusion injury [25–27]. As the core cells of the liver, hepatocytes are the first to be affected in HIRI, suffering direct and severe damage that severely impairs the liver's ability to carry out its normal physiological activities. In contrast, nonparenchymal cells, particularly Kupffer cells, act as detectors and amplifiers of the hepatic inflammatory response, stimulating and promoting the parenchymal cell response through enhanced signaling [28]. Ultimately, the interaction between liver parenchymal and nonparenchymal cells leads to liver injury and apoptosis. This process involves complex intercellular communication and immune responses that collectively drive the development of liver injury [29, 30]. Our study focuses on elucidating the mechanisms of parenchymal hepatocyte injury in HIRI, which not only contributes to understanding disease progression but also provides potential targets for future therapeutic strategies. By studying the HL-7702 cell line in detail, we could assess the effects of HIRI on liver function more accurately and explore potential protective measures. In addition, we recognized that the interaction between parenchymal hepatocytes and nonparenchymal cells is critical for liver injury and repair processes. Future studies will be expanded to include Kupffer cells and other cell types to understand intercellular communication and immune responses in HIRI fully.

In this study, we used 6 h, 12 h, and 24 h as key time points to investigate the response of hepatocytes during HIRI. This choice was based on a comprehensive review of existing literature, where we found that although previous studies have used different time points to construct HIRI models, such as 6 h hypoxia followed by 12 h reoxygenation for the RAW 264. 7 cell line [25, 31], 12 h hypoxia followed by 12 h reoxygenation for the AML cell line [14], using different reoxygenation time points after 6 h hypoxia [26], and for the HL-7702 cell line with 4 h hypoxia followed by 12 h reoxygenation [13], as well as primary hepatocytes with 6 h hypoxia followed by reoxygenation at different time points [21], studies for medium and long periods, especially 12 and 24 h, are still lacking [32]. This inspired us to investigate these understudied time points to reveal the cellular response to HIRI at different stages. Secondly, the time points

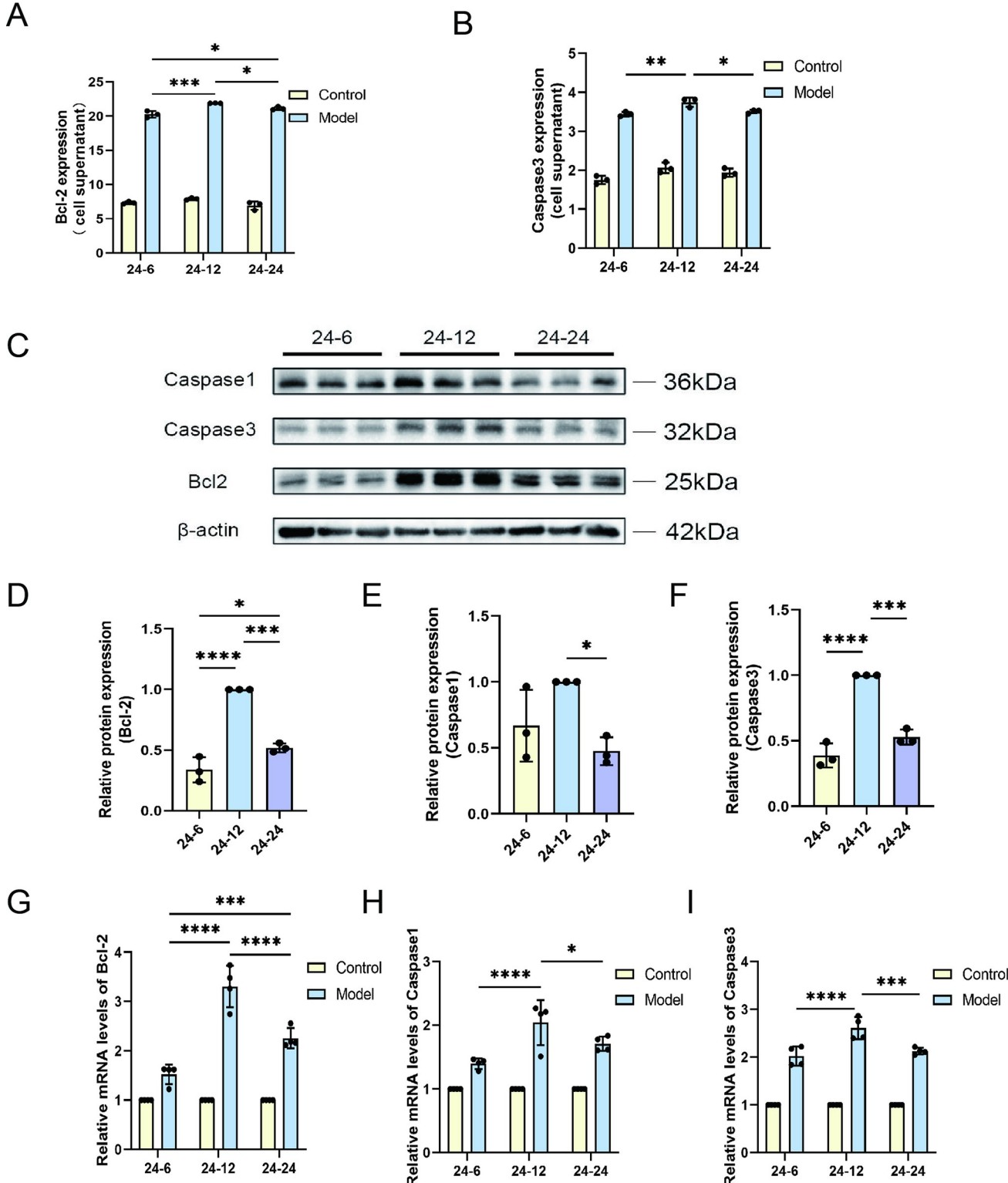

**Fig 5. Expression levels of apoptosis and pyroptosis markers in HL-7702 cells following hypoxia at various reoxygenation durations.** (A) Bcl2 and (B) Caspase3 levels were measured in cells using ELISA, with each condition assessed in triplicate (n = 3). (C) Protein expression levels of Bcl2, Caspase3, and Caspase1 were evaluated utilizing the technique of Western blotting, and adjusted in relation to the internal control gene β-actin within the cells. The comparative levels of (D) Bcl2, (E) Caspase1, and (F) Caspase3 proteins was quantified by normalizing to the 24-hour hypoxia and 12-hour reoxygenation group (24–12), serving as a comparative benchmark (n = 3). QRT-PCR was utilized to evaluate the mRNA levels of (G) Bcl-2, (H) Caspase1, and (I)

Caspase3 (n = 4). Data are depicted as mean values accompanied by SEM, and the statistical significance is determined. (*P<0.05, **P<0.01, ***P<0.001, ****P<0.0001).

we chose were intended to reflect the various stages of the biological response of cells to hypoxia and reperfusion. The 6-hour time point represents the early cellular response to injury, which may involve initial stress signaling and activation of injury detection pathways. The 12-hour time point may reveal the progression of cellular injury and early signs of cell death, a critical turning point at which the cell may begin to recover from injury or further deterioration. The 24-hour time point, on the other hand, may indicate long-term effects of injury and peak of cell death.

The objective of this study was to investigate how varying durations of hypoxia impact cellular models of HIRI. In this study, we investigated the impact of varying hypoxia times (6 h, 12 h, and 24 h) on cells by establishing a cellular IRI model. We reoxygenated the cells for 6 h, 12 h, and 24 h after determining the hypoxia time, respectively, by detecting the inflammatory factors TNF-α, IL-1β, IL-6, as well as apoptosis-related indicators Bcl2, Caspase3, and cellular focal indicator Caspase1.

Firstly, we found that hypoxia time was a key factor affecting the cellular HIRI model. The degree of cellular damage gradually increased with the prolongation of hypoxia time. Especially under 24 hours of hypoxia, expression of intracellular inflammatory factors and apoptosis-related proteins was significantly upregulated, indicating that the cells were subjected to severe metabolic and oxidative stress. In HIRI, the activation of immune cells and the release of inflammatory mediators showed prominent time-dependent characteristics. Inflammatory response gradually increased with the prolongation of ischemic time, leading to more significant cellular damage and dysfunction [33–36]. It was noted that in the animal model, the damage to hepatocytes was proportional to the duration of ischemia [37]. Our cellular experimental data further corroborate this theory.

In addition, reoxygenation phase is a critical period leading to increased cellular damage during HIRI. In our study, we found that after 24 hours of hypoxia, 12-hour reoxygenation was sufficient to trigger a significant inflammatory response and apoptosis, whereas 24-hour reoxygenation was observed as a decrease in apoptosis and expression of inflammatory factors, which may be related to adaptive changes in cells or post-apoptotic clearance mechanisms. During reperfusion, despite improved oxygen supply, mitochondrial problems and activation of neutrophils may trigger inflammation and penetrate nuclear membrane, causing DNA damage and, ultimately, apoptosis [38, 39]. Based on clinical observations, it has been found that within 6 hours after the onset of HIRI, markers of AST and ALT rise significantly to a maximum point, after which these markers begin to decline slowly and return to the normal range after 7 days [40]. And some other studies found that AST and ALT peaked after 1 day [41]. Since the in vivo environment is more complex and involves multiple cell types and physiological factors, the *in vitro* environment is relatively simplified. It may affect the cellular response time to injury and recovery.

Although the present study provides essential information about cellular HIRI modeling, some things could be improved. For example, our experiments were conducted under in vitro conditions that may not fully mimic the complex environment in vivo. Future studies will consider validating these findings in an in vivo model and exploring potential therapeutic interventions. In addition, the detection of cleaved Caspase3 and other members of the caspase family was not included in this study. In future studies, we plan to include the detection of these critical apoptotic markers to provide a more comprehensive analysis of apoptosis. This

will improve our understanding of the mechanisms of HIRI-induced cell death and may reveal new therapeutic targets.

## Conclusion

In summary, the present study reveals significant effects of hypoxia and reoxygenation on cellular inflammation, apoptosis, and pyroptosis pathways, emphasizing the importance of time-dependent responses in pathological processes. In addition, our study provides new insights into the effects of different hypoxia and reoxygenation times on cellular HIRI. These findings enhance our understanding of the biological processes involved and provide an essential basis for developing future therapeutic strategies.

## Supporting information

**S1 Raw images.**
(DOCX)

**S1 Data.**
(ZIP)

## Acknowledgments

We express our heartfelt gratitude to all members of our research group for their invaluable help and unwavering support.

## Author Contributions

**Conceptualization:** Xinlu Xu, Tanfang Zhou, Yingmei Shao.

**Data curation:** Xinlu Xu, Tanfang Zhou, Alimu Tulahong.

**Formal analysis:** Xinlu Xu, Tanfang Zhou.

**Project administration:** Rexiati Ruze, Yingmei Shao.

**Supervision:** Yingmei Shao.

**Writing – original draft:** Xinlu Xu.

**Writing – review & editing:** Tanfang Zhou, Alimu Tulahong, Rexiati Ruze, Yingmei Shao.

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
