## [Decision Letter · Decision Letter 0]

14 Aug 2024

PONE-D-24-29139Exploring the effects of hypoxia and reoxygenation time on hepatocyte apoptosis and inflammationPLOS ONE

Dear Dr. Shao,

Thank you for submitting your manuscript to PLOS ONE. After careful consideration, we feel that it has merit but does not fully meet PLOS ONE’s publication criteria as it currently stands. Therefore, we invite you to submit a revised version of the manuscript that addresses the points raised during the review process.

 Please submit your revised manuscript by Sep 28 2024 11:59PM. If you will need more time than this to complete your revisions, please reply to this message or contact the journal office at plosone@plos.org. Please include the following items when submitting your revised manuscript:A rebuttal letter that responds to each point raised by the academic editor and reviewer(s). You should upload this letter as a separate file labeled 'Response to Reviewers'.A marked-up copy of your manuscript that highlights changes made to the original version. You should upload this as a separate file labeled 'Revised Manuscript with Track Changes'.An unmarked version of your revised paper without tracked changes. You should upload this as a separate file labeled 'Manuscript'.

We look forward to receiving your revised manuscript.

Kind regards,

Manisha Nigam

Academic Editor

PLOS ONE

Journal Requirements:

   "the Open Project Fund for the State Key Laboratory of Central Asian High Disease Pathogenesis and Prevention (SKL-HIDCA-2023-2); the National Natural Science Foundation of China (82360111)"

5. Please remove your figures from within your manuscript file, leaving only the individual TIFF/EPS image files, uploaded separately. These will be automatically included in the reviewers’ PDF.

6. Please include your tables as part of your main manuscript and remove the individual files. Please note that supplementary tables (should remain/ be uploaded) as separate "supporting information" files

7. PLOS ONE now requires that authors provide the original uncropped and unadjusted images underlying all blot or gel results reported in a submission’s figures or Supporting Information files. This policy and the journal’s other requirements for blot/gel reporting and figure preparation are described in detail at https://journals.plos.org/plosone/s/figures#loc-blot-and-gel-reporting-requirements and https://journals.plos.org/plosone/s/figures#loc-preparing-figures-from-image-files. When you submit your revised manuscript, please ensure that your figures adhere fully to these guidelines and provide the original underlying images for all blot or gel data reported in your submission. See the following link for instructions on providing the original image data: https://journals.plos.org/plosone/s/figures#loc-original-images-for-blots-and-gels.   

Additional Editor Comments:

After careful consideration and thorough peer review, we have decided to offer you the opportunity to revise your manuscript for potential publication. The reviewers have provided detailed feedback, and while they recognize the value and potential impact of your work, they have identified several areas that require substantial revisions.

Reviewers' comments:

Reviewer's Responses to Questions

**Comments to the Author**

1. Is the manuscript technically sound, and do the data support the conclusions?

Reviewer #1: Yes

Reviewer #2: Partly

2. Has the statistical analysis been performed appropriately and rigorously? 

Reviewer #1: Yes

Reviewer #2: I Don't Know

3. Have the authors made all data underlying the findings in their manuscript fully available?

Reviewer #1: Yes

Reviewer #2: Yes

4. Is the manuscript presented in an intelligible fashion and written in standard English?

Reviewer #1: Yes

Reviewer #2: No

5. Review Comments to the Author

Reviewer #1: The present study is very interesting. It is well known that IRI of solid organs is a major problem, which seriously damages human health. IRI is an unavoidable pathological process in transplantation and many surgeries, which contain complex molecular events such as pyroptosis, apoptosis, inflammasome and so on. HR of hepatocytes mimics liver IRI, which is very common in liver transplantation and liver resection.

1. 4. It is very important in liver IRI to distinguish the target cells of function. Previous reports have demonstrated that Kupffer cells and parenchymal hepatic cells are both important in liver IRI. What is the main reason for choosing AML instead of KCs or other cell types? Please quote and add to the discussion: PMID: 34217994, PMID: 36058783.

2. We noticed that cleaved-caspase3 and 1 in figure3 were not tested. Is it necessary to test them? If the authors are indeed unable to detect, it is hoped that this item will be added to the discussion as a limitation.

3. This article takes a very interesting perspective, focusing not on upstream molecules but on the model itself. We have some questions. Some common time points in previous liver IRI and HR reports did not appear in this paper. Please explain how the authors considered the selection of time points. Please quote and add to the discussion: PMID: 32532961, 35131594, 30686117.

4. On what basis did the authors assess the state of the cells under the microscope?

Reviewer #2: I have reviewed the manuscript by Shao et al., titled "Exploring the Effects of Hypoxia and Reoxygenation Time on Hepatocyte Apoptosis and Inflammation." This study aims to develop an in vitro model for Hepatic Ischemia-Reperfusion Injury (HIRI), a critical concern during liver surgeries such as transplantation and resection. The researchers utilized the human liver cell line HL-7702 to simulate HIRI, employing a hypoxia chamber to mimic ischemic conditions followed by reoxygenation. They observed that inflammation and apoptosis levels in the cells increased with prolonged hypoxia (6, 12, 24 hours), reaching a peak at 24 hours when coupled with a fixed 12-hour reoxygenation period.

1. The authors mention that the HL-7702 cell line, a typical human liver cell line, was preserved in the lab and revived. The specific passage number of these cells should be included for clarity and reproducibility.

2. The rationale for selecting the HL-7702 cell line warrants further explanation. This cell line may not fully capture the complex interactions of various liver cell types (e.g., hepatocytes, Kupffer cells, endothelial cells) and the broader physiological context of HIRI in humans.

3. The method used to confirm cellular hypoxia after exposing the cells to a gas mixture containing 95% N2 and 5% CO2 should be detailed. How was successful hypoxia induction verified in these cells?

4. In Figure 1, the morphology of hypoxic and normoxic cells is presented at different time intervals. What observations were made regarding the control (normoxic) cells during these intervals? The images suggest the control cells were fully confluent, and the depicted morphological changes might be common regardless of oxygenation conditions. Were specific biomarkers used to validate these morphological observations?

5. In Figure 2, the expression of TNF-alpha via Western blotting appears inconsistent across different time points. Could the authors comment on this finding, especially in the context of expected variations with time?

6. The justification for using two different housekeeping proteins in Figures 2 and 3 during Western blotting should be provided. What was the rationale behind this choice?

7. Figure 4: Western blotting experiments show signs of improper protein loading. The figure might benefit from adjustments to improve readability and accuracy.

8. While the study emphasizes inflammation and apoptosis as key markers of HIRI, other crucial processes such as oxidative stress, autophagy, and mitochondrial dysfunction might also play significant roles. The authors could consider discussing these aspects to provide a more comprehensive view of HIRI mechanisms.

9. Significant improvements in English and corrections of typographical errors are required throughout the manuscript.

6. PLOS authors have the option to publish the peer review history of their article (what does this mean?). If published, this will include your full peer review and any attached files.

Reviewer #1: No

Reviewer #2: **Yes: **Gaurav Joshi

---

## [Author Response · Author response to Decision Letter 0]

30 Aug 2024

Comment from Reviewer #1: 

[1. It is very important in liver IRI to distinguish the target cells of function. Previous reports have demonstrated that Kupffer cells and parenchymal hepatic cells are both important in liver IRI. What is the main reason for choosing AML instead of KCs or other cell types? Please quote and add to the discussion: PMID: 34217994, PMID: 36058783.]

Response 1: First, thank you for your interest in our study and for raising important questions about cell type selection. In response to your request, we have elaborated in the Discussion section and cited references to the literature you provided and other relevant literature to support our view. Although Kupffer cells and other nonparenchymal cells play a vital role in the inflammatory response and immune regulation in liver ischemia-reperfusion injury, the primary reason we chose to study the HL-7702 cell line was based on its representativeness as parenchymal hepatocytes. Parenchymal hepatocytes are the primary functional cells of the liver and are directly involved in critical physiological processes such as metabolism, detoxification and regeneration. In HIRI, these cells bear the brunt of direct and severe injury, which significantly affects the normal physiological functions of the liver. Our study elucidates the response and injury mechanisms of parenchymal hepatocytes in HIRI while recognizing nonparenchymal cells' role. Interactions between parenchymal and nonparenchymal cells collectively drive liver injury through complex intercellular communication and immune responses. We have added the above points in the Discussion section and discussed the different roles of parenchymal and nonparenchymal hepatocytes in HIRI and how they work together to influence liver injury and repair processes. We thank you for your valuable comments and await further guidance.Please find the detailed changes on page 11, lines 286-305.

[2.We noticed that cleaved-caspase3 and 1 in figure3 were not tested. Is it necessary to test them? If the authors are indeed unable to detect, it is hoped that this item will be added to the discussion as a limitation.]

Response 2: First of all, we would like to thank you for your review and valuable comments on our study. We have taken note of your question regarding the untested cleaved-caspase3 and 1 in Figure 3. In future studies, we plan to include cleaved-caspase3 and 1 assay to provide a more comprehensive analysis of apoptosis. We will include this limitation in the Discussion section and discuss how it may affect the results of the study. We believe this will improve our study and contribute more accurately to the scientific discussion in this area. Thank you again for your valuable comments.Please find the detailed changes on page 13, lines 361-366.

[3.This article takes a very interesting perspective, focusing not on upstream molecules but on the model itself. We have some questions. Some common time points in previous liver IRI and HR reports did not appear in this paper. Please explain how the authors considered the selection of time points. Please quote and add to the discussion: PMID: 32532961, 35131594, 30686117.]

Response 3: First, thank you for your high regard for our study and your concern about the time point we chose. In the Discussion section, we will include the literature you recommended and clarify the reasons and rationale for the selected time point. Thank you for your questions, which allowed us to clarify our choice of time point further and emphasize our study's scientific basis and rationale. Please find the detailed changes on page 12, lines 306-324.

[4. On what basis did the authors assess the state of the cells under the microscope?]

Response 4: I appreciate your interest in our study and your inquiry about our methods for assessing the state of cells under the microscope. We carefully observed cell morphology, including cell density and fusion, organelle structure changes, and cellular debris and membrane vesicles, phenomena commonly associated with cellular damage and apoptotic processes. In addition to morphological assessment, we also determined biochemical, inflammatory and apoptotic indicators. Changes in these indicators provided us with a quantitative evaluation of the extent of cellular damage and corroborated with morphological observations. This multifaceted methodology offers a solid basis for assessing cellular damage and increases the reliability of our findings. Thank you again for your valuable comments; we have added and expanded on these aspects in the Results section and look forward to your comments. Please find the detailed changes on page 6, lines 152-165.

Comment from Reviewer #2: 

[1. The authors mention that the HL-7702 cell line, a typical human liver cell line, was preserved in the lab and revived. The specific passage number of these cells should be included for clarity and reproducibility.]

Response 1: First, we would like to sincerely thank you for reviewing our manuscript and providing valuable comments. We fully agree that these points need to be improved. We look forward to addressing them in the revised manuscript. Please find the detailed changes on page 3, lines 82-88.

[2. The rationale for selecting the HL-7702 cell line warrants further explanation. This cell line may not fully capture the complex interactions of various liver cell types (e.g., hepatocytes, Kupffer cells, endothelial cells) and the broader physiological context of HIRI in humans.]

Response 2: We sincerely appreciate your interest in our study and valuable comments on the rationale for selecting the HL-7702 cell line. A single cell line may not be able to fully mimic the complex interactions of different cell types in the liver and the broad physiological context of human HIRI. The following is a description of our rationale for selecting the HL-7702 cell line and its applicability in this study: Hepatocytes, as the significant functional cells of the liver, are directly involved in critical processes such as metabolism, detoxification and regeneration of the liver. In HIRI, hepatocytes are the cell type directly damaged, and the extent of their damage is critical for liver function. This study aimed to investigate the mechanisms of hepatocyte response and injury in HIRI, particularly the changes at different time points. The HL-7702 cell line provided a homogeneous model system that allowed us to focus on hepatocyte behavior without interference from other cell types. We recognize that using a single cell line limits our understanding of the role of non-parenchymal cells such as Kupffer cells and endothelial cells in HIRI. However, the uniqueness of the HL-7702 cell line also allowed us to more precisely assess the response of hepatocytes to hypoxia and reperfusion, providing fundamental data for future precision intervention strategies targeting hepatocytes. Based on the results of the HL-7702 cell line, we plan to introduce more cell types in our future studies to construct more complex liver disease models and more comprehensively mimic the pathological process of HIRI. By focusing on hepatocytes, our study contributes to understanding crucial cell death and inflammatory pathways in HIRI. It provides a scientific basis for the development of novel therapeutic strategies. In the Discussion section, we have further elaborated the rationale for selecting the HL-7702 cell line and discussed its limitations and directions. Please find the detailed changes on page 12, lines 306-324.

[3. The method used to confirm cellular hypoxia after exposing the cells to a gas mixture containing 95% N2 and 5% CO2 should be detailed. How was successful hypoxia induction verified in these cells?]

Response 3: Thank you for your interest in our study and for pointing out the importance of validating cellular hypoxia status. We have elaborated our method to ensure the accuracy and reproducibility of cellular hypoxia. Your valuable comments are appreciated and will be improved in the revised manuscript. Please find the detailed changes on page 4, lines 90-104.

[4. In Figure 1, the morphology of hypoxic and normoxic cells is presented at different time intervals. What observations were made regarding the control (normoxic) cells during these intervals? The images suggest the control cells were fully confluent, and the depicted morphological changes might be common regardless of oxygenation conditions. Were specific biomarkers used to validate these morphological observations?]

Response 4: First of all, thank you for your interest in our study and your questions about the results of our experiments. In our experiments, control cells (normoxic conditions) were observed at the same time intervals. We recorded the morphological changes of these cells, including cell fusion, to ensure that we could accurately compare the cellular responses under hypoxic and normoxic conditions. Through microscopic observation, we noted that the control cells gradually fused during the culture, a common phenomenon in cell culture. We ensured that this fusion did not affect the assessment of cell damage under hypoxic conditions. We performed inflammation and apoptosis-related experiments to further validate the morphological observations under the microscope. The changes in these biomarkers were consistent with the morphological changes we observed in the cells, confirming the biological significance of cell injury under hypoxic conditions. Thank you for your valuable comments; we added and expanded these aspects in the results section.Please find the detailed changes on page 6, lines 152-165.

[5. In Figure 2, the expression of TNF-alpha via Western blotting appears inconsistent across different time points. Could the authors comment on this finding, especially in the context of expected variations with time?]

Response 5: First, we would like to express our sincere appreciation for your interest in our study data. We have noted the changes in TNF-alpha expression at different time points in Figure 2 and have considered these observations in our analysis. Although changes in expression existed, they did not show statistical significance, possibly due to limited sample size, biological variability, or other experimental factors. The limitations of this observation may have had some impact on the complete interpretation of the data. However, the overall effect of these changes on the study's conclusions is minor. To overcome these limitations and provide deeper insights into future studies, we plan to increase the sample size and employ more stringent experimental conditions to control for possible variables. With these measures, our study will be strengthened and make a more valuable contribution to the scientific discussion in this field. We thank you for your valuable comments and will fully consider them in the revised manuscript.

[6. The justification for using two different housekeeping proteins in Figures 2 and 3 during Western blotting should be provided. What was the rationale behind this choice?]

Response 6: First, we thank you for your interest in our study and for pointing out the problem of using different housekeeping proteins in Western blotting experiments. Our choice was based on experimental design considerations and the pursuit of clarity in the experimental results. In the experiment in Figure 3, the expected molecular mass of Caspase1 is about 36 kDa, which is very close to the 38 kDa of GAPDH. Since the molecular weights of these two proteins are very near and under our experimental conditions, their migration distances on the gel may be difficult to distinguish. To avoid this confusion, we chose β-actin as the housekeeping protein in Figure 3, which has a molecular weight of 42 kDa and is more clearly distinguishable from Caspase1 and GAPDH. We believe that using β-actin as a housekeeping protein can more accurately assess the expression level of Caspase1 while maintaining clarity and reproducibility of the results. This choice helps to ensure the accuracy and reliability of our Western blotting results, thus providing a solid foundation for the study conclusions.

[7. Figure 4: Western blotting experiments show signs of improper protein loading. The figure might benefit from adjustments to improve readability and accuracy.]

Response 7: We want to express our sincere appreciation for your review of our manuscript and your valuable comments. We fully agree that these issues need to be improved. We look forward to addressing them in the revised manuscript.

[8. While the study emphasizes inflammation and apoptosis as key markers of HIRI, other crucial processes such as oxidative stress, autophagy, and mitochondrial dysfunction might also play significant roles. The authors could consider discussing these aspects to provide a more comprehensive view of HIRI mechanisms.]

Response 8: We are grateful for your valuable suggestions. What you have pointed out about key processes other than inflammation and apoptosis play an essential role in the development of HIRI. We have added and expanded on these aspects in the discussion section.Please find the detailed changes on page 11, lines 274-285.

[9. Significant improvements in English and corrections of typographical errors are required throughout the manuscript.]

Response 9: First, thank you for reviewing our manuscript and your valuable comments. We apologize for the English and typographical errors in the manuscript and fully agree that these issues need to be improved. Thank you again for your valuable time and suggestions, and we look forward to addressing these issues in the revised manuscript.

---

## [Editor Report · Decision Letter 1]

4 Sep 2024

Exploring the effects of hypoxia and reoxygenation time on hepatocyte apoptosis and inflammation

PONE-D-24-29139R1

Dear Dr. Shao,

We’re pleased to inform you that your manuscript has been judged scientifically suitable for publication and will be formally accepted for publication once it meets all outstanding technical requirements.

Kind regards,

Manisha Nigam

Academic Editor

PLOS ONE

Additional Editor Comments (optional):

I am pleased to write that in your manuscript titled "Exploring the effects of hypoxia and reoxygenation time on hepatocyte apoptosis and inflammation" you have successfully addressed the reviewers’ comments, and the quality and significance of your work are evident. I therefore recommend this manuscript for the publication.
---

## [Editor Report · Acceptance letter]

12 Oct 2024

PONE-D-24-29139R1 

PLOS ONE

Dear Dr. Shao, 

I'm pleased to inform you that your manuscript has been deemed suitable for publication in PLOS ONE. Congratulations! Your manuscript is now being handed over to our production team.

Kind regards, 

on behalf of

Dr. Manisha Nigam 

Academic Editor

PLOS ONE